# A C-terminal peptide from type I interferon protects the retina in a mouse model of autoimmune uveitis

**Chulbul M. Ahmed[1], Cristhian J. Ildefonso[2], Howard M. Johnson[3], Alfred S. Lewin[1]***

**1** Department of Molecular Genetics and Microbiology, University of Florida, Gainesville, FL, United States of America, **2** Department of Ophthalmology, University of Florida, Gainesville, FL, United States of America, **3** Department of Microbiology and Cell Science, University of Florida, Gainesville, FL, United States of America

* lewin@ufl.edu

**Data Availability Statement:** The data underlying this study have been uploaded to the Harvard Dataverse and are accessible using the following URL: https://dataverse.harvard.edu/dataset.xhtml?

## Abstract

Experimental autoimmune uveitis (EAU) in rodents recapitulates many features of the disease in humans and has served as a useful tool for the development of therapeutics. A peptide from C-terminus of interferon α1, conjugated to palmitoyl-lysine for cell penetration, denoted as IFNα-C, was tested for its anti-inflammatory properties in ARPE-19 cells, followed by testing in a mouse model of EAU. Treatment with IFNα-C and evaluation by RT-qPCR showed the induction of anti-inflammatory cytokines and chemokine. Inflammatory markers induced by treatment with TNFα were suppressed when IFNα-C was simultaneously present. TNF-α mediated induction of NF-κB and signaling by IL-17A were attenuated by IFNα-C. Differentiated ARPE-19 cells were treated with TNFα in the presence or absence IFNα-C and analyzed by immmunhistochemistry. IFNα-C protected against the disruption integrity of tight junction proteins. Similarly, loss of transepithelial resistance caused by TNFα was prevented by IFNα-C. B10.RIII mice were immunized with a peptide from interphotoreceptor binding protein (IRBP) and treated by gavage with IFNα-C. Development of uveitis was monitored by histology, fundoscopy, SD-OCT, and ERG. Treatment with IFNα-C prevented uveitis in mice immunized with the IRBP peptide. Splenocytes isolated from mice with ongoing EAU exhibited antigen-specific T cell proliferation that was inhibited in the presence of IFNα-C. IFNα-C peptide exhibits anti-inflammatory properties and protects mice against damage to retinal structure and function suggesting that it has therapeutic potential for the treatment of autoimmune uveitis.

## Introduction

Uveitis refers to an inflammation affecting the uveal and retinal layers of the eye that accounts for 10–15% of blindness in US[1]. In rodent models of the disease, histological analysis shows injury to retinal layers, retinal folds, subretinal fluid, damage to retinal blood vessels and the retinal pigment epithelium (RPE), and choroiditis. Uveitis may affect both the anterior and posterior chambers of the eye [2,3]. Left untreated, it can lead to permanent retinal damage, cystoid macular edema and vitreous opacity[4]. Uveitis can arise from an infection, systemic inflammation, or from a local autoimmune response. It may also be a component of a systemic

persistentId=doi:10.7910/DVN/IZE4OA. The DOI for access is: 10.7910/DVN/IZE4OA.

**Funding:** ASL Shaler Richardson Endowment https://research.ufl.edu/ufrf.html. The sponsors played no role in the study design.

**Competing interests:** The University of Florida been awarded a patent (US 9,951,111B2) governing the interferon peptide mimetic used in this study. Two of the authors (CMA and HMJ) may obtain royalties from the licensing of this technology. This does not alter our adherence to PLOS ONE policies on sharing data and materials.

autoimmune response involving multiple organs. Several conditions can cause uveitis secondarily, including sarcoidosis [5], Behcet's disease [6], and Vogt-Koyanagi-Harada (VKH) syndrome [7]. Autoimmune disorders such as Crohn's disease [8], rheumatoid arthritis [9], and multiple sclerosis (MS) [10,11], can also have an overlapping uveitis. After the initial trigger, the immunogenic pathways in these different diseases share many common features. Both the innate and adaptive immune responses are activated in autoimmune uveitis.[12,13]. An animal model of the disease, experimental autoimmune uveitis (EAU) is established by inoculating mice with specific antigens including peptides derived from interphotoreceptor retinoid binding protein (IRBP), RPE65, rhodopsin, retinal arrestin, recoverin, and phosducin. Transgenic lines of mice have been established that express a T cell receptor that is specific for immunogenic peptides derived from IRBP, and one of these lines develops uveitis spontaneously, providing a model that avoids the need for immunization [14]. Uveitis may also be induced by adoptive transfer of uveitic antigen-specific T cells [2]. Such models have been important in deciphering the mechanisms of disease and in testing therapeutic approaches.

Aside from external factors, the host microbiome has been identified in recent years as a contributing factor to both infectious and autoimmune uveitis. Commensal microbes from intestine, saliva, or ocular source have been implicated in the initiation of uveitis [15–17]. It is believed that through a process involving antigen mimicry followed by an adjuvant effect, these microbes can produce autoreactive lymphocytes leading to the development of uveitis. Thus, what is gleaned from rodent models of the disease can have greater relevance in alleviating the disease arising from host-microbiome interaction.

Non-infectious uveitis is currently being treated by using corticosteroids, general immunosuppressants, or specific antibodies [18,19]. While these treatments provide temporary relief, they are associated with side effects that include cataract formation, increased susceptibility to infection, and elevated intraocular pressure [20]. There is thus a critical need for treatments that are sustained and safer.

The ability of type I interferons (IFNα, IFNβ, IFNτ) to suppress autoimmune and inflammatory responses is well documented [21–23]. This has been exploited to develop treatments for multiple sclerosis [24], psoriasis [25], and coeliac diseases [26]. Several studies have reported the success of use of IFNα in a number of ocular disorders, including uveitis [27–29]. Type I interferon (IFNα or IFNβ) are commonly used to treat relapsing multiple sclerosis (MS) [24]. Nevertheless, its use is associated with lymphopenia, flu-like symptoms, depression, and weight loss. To circumvent these undesirable effects, we have developed a C-terminal peptide of IFNα(residues 152–189), which upon conjugation with palmitoyl-lysine can penetrate the plasma membrane. The resulting modified peptide is denoted as IFNα-C. IFNα-C can recapitulate nearly all the activities of intact IFN [30], including protection against the severe remitting/relapsing paralysis in a mouse model of multiple sclerosis (experimental allergic encephalomyelitis or EAE). Between the regions 152–189, there is 70% homology and 90% conservation between human and mouse IFNα1 sequences. Of note, the IFNα-C peptide lacks the toxicity associated with the parent IFN [30], and may thus represent an improved alternative to parent IFN. Since MS and autoimmune uveitis share common characteristics, we were interested in testing the therapeutic efficacy of IFNα-C peptide in experimental autoimmune uveitis (EAU). Herein, we provide evidence that oral administration of IFNα-C can prevent the intraocular damage associated with EAU.

## Materials and methods

### Peptide synthesis

Human lipo-IFNα1 (152–189) peptide was synthesized by GenScript (Piscataway, N.J.). Lipo refers to conjugation with the lipid moiety, palmitoyl-lysine that was conjugated to the N-terminus of the peptide to allow it to penetrate the plasma membrane. The resulting peptide was dissolved in DMSO at 10 mg/ml. Further dilutions to the desired concentration were carried out in PBS under sterile conditions, as described before [30]. In control cells, DMSO at the concentration present in the diluted IFN peptide was added to the cells and is indicated as vehicle control.

### Cell culture

ARPE-19 cells were obtained from ATCC (Manassas, VA) and propagated in DMEM/F-12 medium containing 10% FBS and 1% each of Penicillin, Streptomycin in a humidified incubator at 37°C.

### Relative quantitation of mRNA expression by RNA extraction and qPCR

ARPE-19 cells were grown overnight in a 12 well plate seeded at 70% confluence. The next day, the medium was changed to DMEM/F-12 containing low FBS (1%). Cells were pre-treated with 3 μM IFNα-C for 4 hr. Cells were then washed with PBS and total RNA was extracted using the RNeasy Mini Kit from QIAGEN, as described before [31]. One microgram of RNA was used to synthesize first strand cDNA using the iScript cDNA synthesis kit from Bio-Rad (Hercules, CA). The primers used for qPCR were synthesized by IDT (Coralville, Iowa) and are listed in **Table 1**.

**Table 1. Sequence of primers used for qPCR.**

| Name | Sequence (5'-3') |
|---|---|
| TGFβ F | AGCGACTCGCCAGAGTGGTTA |
| TGFβ R | GCAGTGTGTTATCCCTGCTGTCA |
| TWST1 F | CGGGAGTCCGCAGTCTTA |
| TWST1 R | CTCTGGAGGACCTGGTAGAG |
| TTP F | CCATCTACGAGAGCCTCCTGT |
| TTP R | AAGTGGGTGAGGGTGACAGCT |
| Foxp3 F | CTGACCAAGGCTTCATCTGTG |
| Foxp3 R | ACTCTGGGAATGTGCTGTTTC |
| IL-1 β F | CTCGCCAGTGAAATGATGGCT |
| IL-1 β R | GTCGGAGATTCGTAGCTGGAT |
| IL-6 F | CTTCTCCACAAGCGCCTTC |
| IL-6 R | CAGGCAACACCAGGAGCA |
| IL-8 F | GCAGCCTTCCTGATTTCTGCA |
| IL-8 R | CCAGACAGAGCTCTCTTCCATCAG |
| CCL-2 F | CTCATAGCAGCCACCTTCATTC |
| CCL-2 R | TCACAGCTTCTTTGGGACACTT |
| β-actin F | AGCGAGCATCCCCCAAAGTT |
| β-actin R | GGGCACGAAGGCTCATCATT |

Forward; R, Reverse; TWST, Twist; TTP, tristetraprolin.

The PCR reaction mixture contained cDNA template, SsoFastEvaGreen Super mix containing SYBR green (Bio-Rad, Hercules, CA), and 3 μM target specific primers. After denaturation at 95˚C for 2 min, 40 cycles of reaction including denaturation at 94˚C for 15 sec followed by annealing at 60˚C for 30 sec were carried out using C1000 thermal cycler CFX96 real-time system (Bio-Rad). Gene expression was normalized to beta actin. Relative gene expression was compared with untreated samples and determined using the CFX96 software from Bio-Rad. To investigate the anti-inflammatory properties of IFNα-C, ARPE-19 cells were seeded as above, and pretreated with IFNα-C (3 μM) for 2 hr, followed by TNFα (50 ng/ml) for 4 hr. RNA extraction and qPCR were carried out, as described above. The primers used are listed in Table 1. In these experiments the Ct value for β-actin in untreated samples was 16.1 with a standard deviation of 0.18, and the Ct values for the cytokine/chemokine analytes ranged between 19.03 (CCL-2 following TNFα treatment) to 33.1 (IL-1β in untreated cells).

## Quantitation of IL-1β using ELISA

ARPE-19 cells were seeded at 70% confluence in 12 well plate and grown overnight. They were then incubated in medium containing 1% FBS and treated with IFNα-C (3 μM) followed by treatment with TNFα (50 ng/ml) for 24 hr. Supernatants were harvested and used for quantitation of IL-1β in an ELISA format, using a kit from BioLegend (San Diego, CA).

## Immunohistochemistry

ARPE-19 cells were seeded in 8 well chambered slides and grown overnight in regular DMEM/F12 medium with 10% FBS. Next day, the medium was changed to DMEM/F12 medium without the serum and pre-treated with IFNα-C (3 μM) for 2 hr followed by addition of TNFα or IL-17A (both at 50 ng/ml) for 30 min. For immunohistochemistry, cells were fixed with 4% paraformaldehyde for 30 min at room temperature and washed with PBS, followed by permeabilization of cells in PBS with 1% Triton X-100 for 30 min at room temperature. Cells were then blocked in 10% normal goat serum in PBS containing 0.5% Triton X-100 for 30 min at room temperature followed by washing in 0.2% Triton X-100 in PBS (wash buffer). Antibody to p65 (cat no. 8242S, Cell Signaling Technology), the active subunit of NF-kB, pSTAT3 (cat no. 9145S, Cell Signaling Technology), or STAT3 (cat no. sc-8019, Santa Cruz Biotechnology) were added to cells and incubated for 2 hr, followed by washing and staining with secondary antibody conjugated with Cy3 (cat no. A10520, Invitrogen) (for p65), or Alexa488 (cat no. A32371, Invitrogen) for pSTAT3 and STAT3, and DAPI for 1 hr. Cells were then washed. For ARPE-19 cells grown for 4 weeks and treatment, antibody to ZO-1 (cat no. 18–7430, Invitrogen) was added and incubated overnight, followed by washing and incubation with Cy-3 conjugated anti-rabbit secondary antibody for 1 hr. After washing, mounting media was added. Cells were covered with a cover slip and imaged in a Keyence BZ-X700 fluorescence microscope.

## Measurement of transepithelial electrical resistance (TEER)

We plated the ARPE19 cells on Transwell inserts (Greiner-Bio-one, surface area 33.6 mm$^2$, 0.4 μm pore size) in DMEM/F-12 with 1% FBS for 4 weeks, with a change of media twice per week in order to obtain a differentiated monolayer characterized by tight junctions and a cobblestone appearance [32]. It should be noted that despite their ability to form tight junctions, ARPE-19 monolayers do not form melanosomes and express low levels of marker proteins such as CRALBP and RPE65 [32,33]. Cells were pre-treated with IFNα-C (3 μM) for 4 hr followed by treatment with TNFα (50 ng/ml) for 48 hr. We used an EVOM2 voltohmmeter (World Precision Instruments, Sarasota, FL) to measure transepithelial electrical resistance.

The inserts were removed one at a time and placed into the EVOM2 chamber filled with DMEM/F-12 basal media. We calculated the net resistance values by subtracting the value of a blank transwell filter from the value of each filter with plated cells. The samples were run in triplicates and the values show mean with standard deviation.

## Experimental autoimmune uveitis and administration of peptide

All procedures were approved by the University of Florida Institutional Animal Care and Use Committee and were conducted in accordance with the Association for Research in Vision and Ophthalmology (ARVO) Statement for the Use of Animals in Ophthalmic and Vision Research. We maintained B10.RIII mice (Jackson Laboratory) in a 12h light: 12 hour dark cycle in a barrier facility. We used B10.RIII mice since they carry H2$^r$ allele of the MHC gene and are more amenable to immunization (2, 3). B10.RIII mice (female, 6 to 8 weeks old, n = 8) were immunized with 100 μg of human IRBP$^{160-181}$ peptide (GenScript, Piscataway, N.J.) dissolved 100 μl PBS emulsified in 100 μl of complete Freund's adjuvant. Mouse numbers were selected based on the effect size expected from our previous experiments with this model [31]. While sex is not reported to influence the susceptibility to disease, female mice have been typically used in this model [3]. We administered the emulsion at the base of the tail and subcutaneously on the thighs. We administered IFNα-C (200 μg/mouse in 200 μl PBS), or PBS (200 μl) by gavage starting two days before injection of IRBP peptide and continuing daily for three weeks after IRBP injection. The peptide dose was based on our previous study in which oral gavage of IFN peptide had protected mice against lethal dose of vaccinia virus [34] Strategies to overcome barriers to oral delivery of peptide drugs has recently been reviewed by Drucker [35].

## Fundoscopy and spectral domain optical coherence tomography (SD-OCT)

These procedures were conducted exactly as described in our previous paper [31]. The health of mice in this study was monitored daily. For these procedures mice were anesthetized with a mixture of ketamine (95 mg/kg) and xylazine (5–10 mg/kg), and at the conclusion of the study mice were euthanized by inhalation of $CO_2$ followed by cervical dislocation. We measured the number of infiltrating cells in the vitreous using Image J software from NIH (URL:https://imagej.nih.gov/ij/), as described before [31]. Briefly, the area representing the vitreous was delineated in the OCT image. The infiltrating cells were converted into binary images and the software quantified the cells in the selected area. We averaged the number of cells in three different B-scans from the stack. We then averaged the number of cells in both eyes from these digital images and averaged.

## Electroretinography (ERG)

Electroretinography measurements in dark adapted mice was conducted as described in Ahmed *et al.* [31].

## Splenocyte proliferation assay

Spleens were harvested from IRBP-immunized B10.RIII mice and treated with IFNα-C, four weeks after immunization. Splenocytes were isolated and seeded at 5 x 10$^5$ cells per well in RPMI medium containing 10% FBS in a 96 well plate and grown in a humidified chamber at 37˚C. IFNα-C (3 μM) was added to cells for 2 hr. IRBP (50 μg/ml) was added to each well and cells were incubated for 72 hr before proliferation was measured using CellTiter 96 Aqueous One cell proliferation assay kit from Promega (Madison, WI). The data represent the average

of 5 different mice immunized and treated with IFNα-C, and are shown as average ± standard deviation.

## Statistical analysis

For comparison of mean transcript levels to untreated cells, we used Student's t test for unpaired samples. In this case, samples were not compared to each other but only to the untreated cells for each group. ERG data was analyzed using GraphPad Prism software (San Diego, CA). Two-way ANOVA followed by a post-hoc Sidak test was used to analyze differences between groups at the same light intensity using an adjusted critical value.

# Results

## IFNα-C induces expression of anti-inflammatory mediators in ARPE-19 cells

We used a spontaneously derived human RPE cell line, ARPE-19 to test the protective effect of IFNα-C. ARPE-19 cells were incubated in the presence or absence of IFNα-C peptide. RNA was extracted from these cells and used to synthesize cDNA and carry out qPCR for target genes. The expression of target genes was normalized to that in untreated cells. Addition of 3 μM IFNα-C to ARPE-19 cells for 4 hrs led to an increase in expression of the anti-inflammatory cytokine, TGFβ (3 fold) (**Table 2**). The anti-inflammatory proteins TWST1 and TTP that cause post-transcriptional downregulation of mRNA of inflammatory mediators, such as TNFα, IL-6, CCL2, and IFNγ (reviewed in [36,37]), were also induced by 3-fold. The Forkhead family transcription factor, Foxp3 that is required for the production of Tregs that are neuroprotective [38] was enhanced by 2-fold. Together, these results point to an anti-inflammatory environment generated in the presence of IFNα-C.

## Immunoprotective effects of IFNα-C in ARPE-19 cells treated with TNFα

Tumor necrosis factor α (TNFα) is associated with the onset of uveitis [39,40]. Therefore, we examined the effect of TNFα treatment (50 ng/ml) on ARPE-19 cells for 4 hrs, and determined whether pre-treatment with the IFNαC (3 μM) for 2 hrs suppressed markers of inflammation (**Table 3**).

RNA extracted from these cells was used to carry out qPCR. Inflammatory cytokines IL-1β, IL-6, and the chemokines IL-8 and CCL-2 (both attract leukocytes) were induced to varying degrees. These cytokines and the chemokines were downregulated by 30–50% by the simultaneous presence of IFNαC, thus suggesting an anti-inflammatory role for IFNαC. The induction of IL-1β and its downregulation by IFNαC was further verified by carrying out ELISA of

**Table 2. IFNα-C added to ARPE-19 cells causes the induction of anti-inflammatory factors.**

| Target gene | Relative expression ± sd | P value |
|---|---|---|
| TGFβ | 3.5 ± 0.6 | 0.001 |
| TWST1 | 3.1 ± 1 | 0.01 |
| TTP | 3.3 ± 0.4 | 0.01 |
| Foxp3 | 2.1 ± 0.6 | 0.01 |

ARPE-19 cells were treated with IFNα-C (3 μM) for 4 hrs, RNA was extracted, used for cDNA synthesis followed by qPCR. Analysis by qPCR for the target genes indicated was carried out using β-actin as a control. Expression in untreated cells was normalized as 1. The results represent the average of three independent experiments (biologic replicates).

**Table 3. IFNα-C suppresses the inflammatory cytokines induced by TNFα treatment.**

| Target gene | Relative expression ± s.d. | | % Reduction | P value |
|---|---|---|---|---|
| | TNFα | TNFα + IFNα-C | | |
| IL-1β | 37 ± 6 | 23 ±2 | 38 | 0.001 |
| IL-6 | 7 ± 0.6 | 3 ± 0.8 | 58 | 0.006 |
| IL-8 | 76 ± 6 | 40 ± 7 | 48 | 0.001 |
| CCL-2 | 36 ± 5 | 26 ± 4 | 28 | 0.005 |

ARPE-19 cells were treated with IFNα-C (3 μM) for 2 hr, followed by treatment with TNF (50 ng/ml) for 4 hr. RNA extraction and qPCR were carried out as in Table 1. Analysis was done using β-actin as a control. The results represent the average of three independent experiments.

the supernatants from these cells (Fig 1). Treatment of ARPE-19 cells with TNFα (50 ng/ml) for 24 hr resulted in the secretion of 250 ± 40 pg/ml of IL-1β, which was reduced to 71 ± 30 pg/ml, by pretreatment of cells with IFNαC (3 μM) for 2 hr followed by treatment with TNFα (50 ng/ml) for 24 hr, while the presence of solvent and TNFα caused a secretion of 210 ± 20 pg/ml of IL-1β, indicating the ability of IFNαC to reduce an inflammatory response.

TNFα-associated inflammation activates the NF-κB promoter. We have determined the effect of IFNα-C on NF-κB promoter by following the nuclear translocation of p65, the active subunit of NF-κB. Addition of TNFα (50 ng/ml) to ARPE-19 cells for 30 min caused translocation of p65 to the nucleus, and translocation was prevented in cells that were pretreated with IFNα-C (Fig 2). IL-17A, the product of T helper 17 cells is a major contributor to the onset of uveitis. Since type I IFN is reported to have suppressive effect on IL-17A [41,42], we wanted to test the effect of IFNα-C on the signaling by IL-17. IL-17 acts through the JAK2/STAT3 pathway that causes the nuclear translocation of STAT3 upon its activation [43,44]

ARPE-19 cells were pretreated with IFNα-C (3 μM for 2 hr) followed by treatment with IL-17A (50 ng/ml) for 30 min (Fig 3). Nuclear translocation of pSTAT3 caused by IL-17A was prevented when IFNα-C was simultaneously present. As a control, staining carried out with antibody to total STAT3 (inactivate form) did not show any nuclear translocation, as reported in our previous publication [31]. Treatment with IFNα-C can thus reduce inflammatory response in ARPE-19 cells.

**Protection against disruption of barrier properties of ARPE-19 cells by IFNαC.** Tight junction proteins provide a functional barrier on the cell surface of RPE cells. ARPE-19 cells were grown in 8 well microscopic slides for 4 weeks in low serum media until they differentiated and acquired a cobblestone appearance similar to primary RPE cells. During uveitis, elevated levels of cytokines such as TNFα or IL-17A cause disruption of RPE barrier properties [39,45]. These were added to differentiated ARPE-19 cells for 48 hr at 50 ng/ml each. Treatment here was for 48 hr as opposed to 24 hr for freshly cultured cells. We treated differentiated ARPE-19 cells for 48 hours, because they are highly resistant to damage. Some of the cells were pre-incubated with IFNα-C (3 μM) for 4 hr prior to the addition of TNFα or IL-17A. To detect the maintenance of cell boundaries, cells were incubated with a primary antibody to zona occludin 1 (ZO-1) (Fig 4).

Cells that were not treated or those treated only with IFNα-C exhibited continuous staining of ZO-1 at the interface of neighboring cells. Treatment with TNFα or IL-17A disrupted this distribution. When IFNα-C was also present in the cells, the cell surface distribution of ZO-1 was maintained. IFNα-C peptide by itself did not have any effect on the distribution of ZO-1.

The integrity of RPE cell junctions can be monitored by measuring the electrical resistance across the monolayer of cells. We measured transepithelial electrical resistance (TEER) in

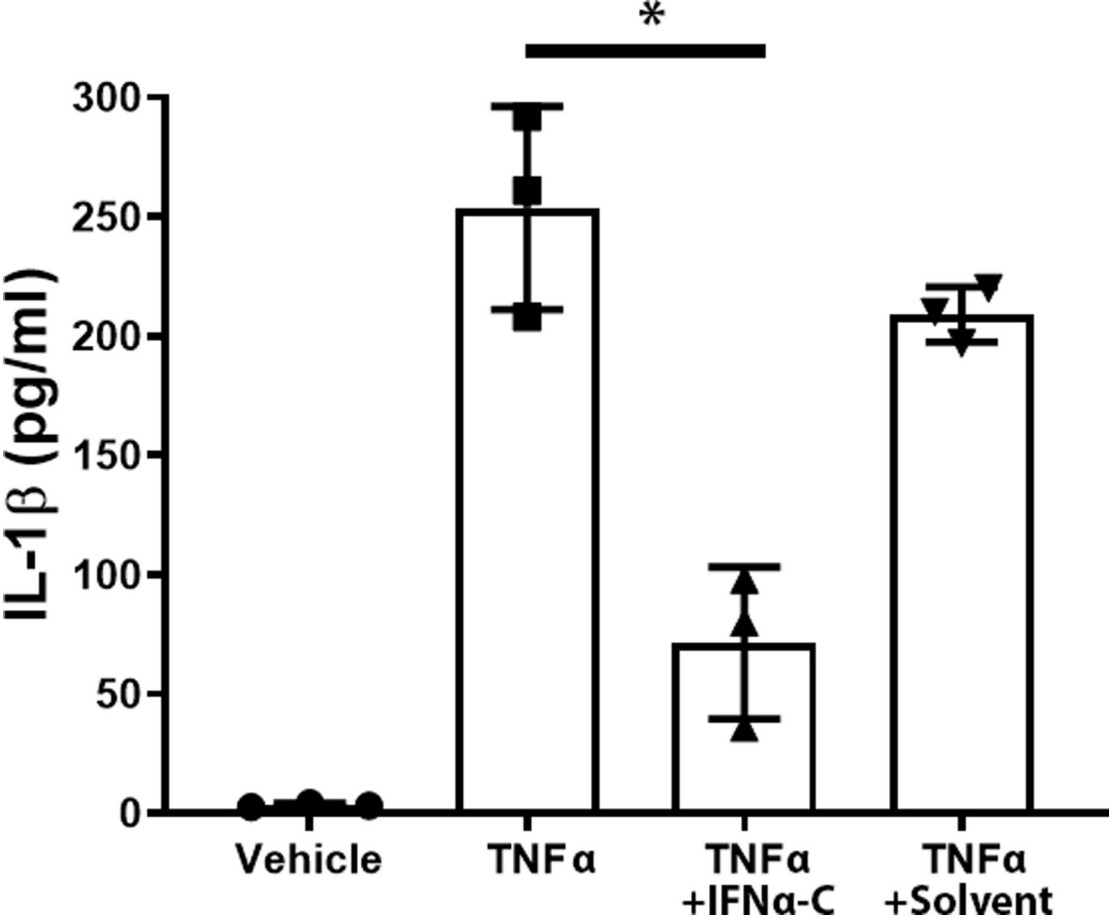

**Fig 1. IFNα-C suppressed the release of IL-1β from ARPE-19 cells induced with TNFα.** ARPE-19 cells were grown overnight in 12-well plates. They were placed in low serum medium and pre-treated with IFNα-C (3 µM) for 2 hr followed by treatment with TNFα (50 ng/ml) for 24 hr. Supernatants were collected and assayed for IL-1β using ELISA. N = 6. The error bars indicate standard deviation. *, P = 0.01.

ARPE-19 cells that had been differentiated by growing them in transwell plates in low serum media for 4 weeks. Cells were incubated with IFNα-C (3 µM) for 4 hr, followed by exposure to TNFα (50 ng/ml) for 48 hr. We measured TEER using a voltohmmeter (**Fig 5**).

In cells without IFNα-C, treatment with TNFα led to a 75% reduction in resistance. When IFNα-C was also present, there was no significant loss in TEER caused by TNFα. This result is consistent with the maintenance of tight junction proteins detected by ZO-1 staining (**Fig 4**). Thus, IFNα-C preserved the distribution and function of tight junction proteins in an inflammatory environment.

## Oral delivery of IFNα-C suppresses autoimmune uveitis in mice

Eight female B10.RIII mice were immunized with IRBP emulsified in complete Freund's adjuvant. Oral gavage with IFNα-C (200 µg in 200 µl PBS per mouse) or with solvent (200 µl PBS per mouse) began on day -2 and was performed daily for 3 weeks after immunization with IRBP peptide. Starting two weeks after immunization, eyes were examined weekly by fundoscopy and SD-OCT. An example of SD-OCT B-scans from day 14 are shown in **Fig 6A**.

Control mice (solvent) showed excessive infiltration of cells in the vitreous, optic nerve head, and the retina. Swelling of the retina and deposits were also noted in control eyes. In

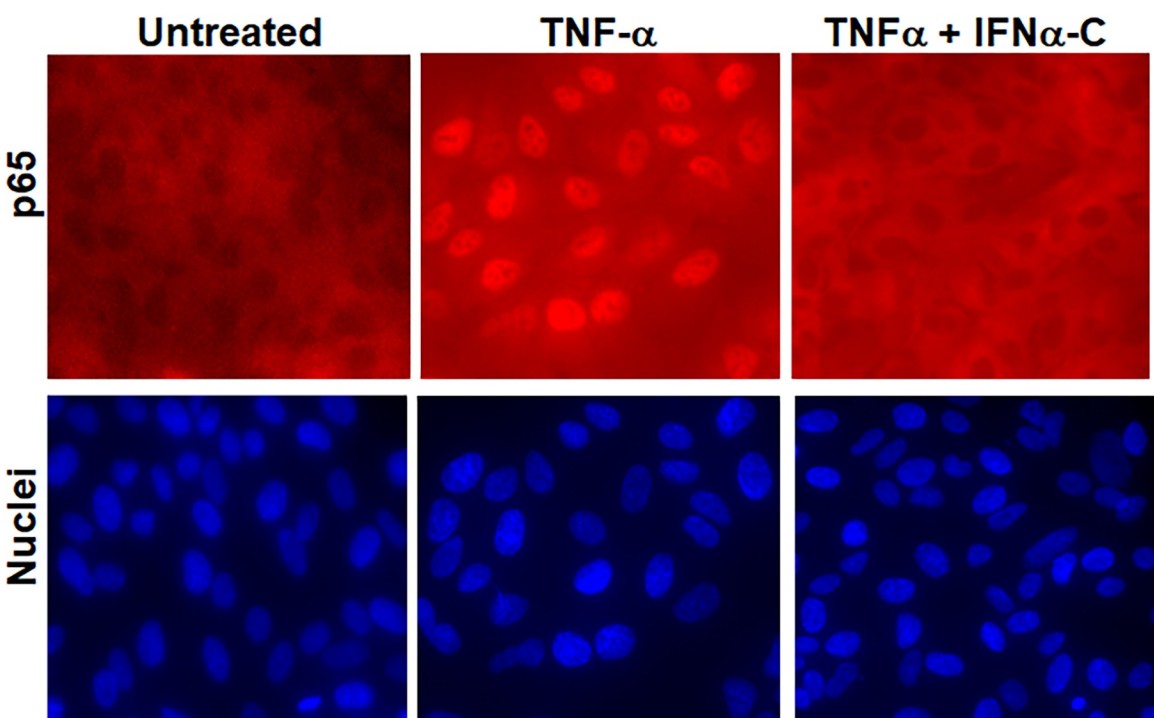

**Fig 2. Suppression of NF-κB signaling by IFNα-C.** ARPE-19 cells were seeded on 8 well microscopic slides and grown overnight. They were taken in low serum medium and treated with IFNα-C (3 μM) for 2 hr followed by treatment with TNFα (50 ng/ml) for 30 min. Cells were fixed, permeabilized with 1% Triton X-100 (in PBS) and stained with an antibody to p65, followed by staining with Cy3-conjugated secondary antibody and DAPI and imaged using fluorescence microscopy at 40x.

contrast, eyes from mice that were orally administered IFNα-C showed significantly fewer infiltrating cells and retinal structures were unaffected. On day 14 after immunization, the average number of inflammatory cells in B-scans from three areas of the posterior chamber in PBS treated mice was 70 ± 10, while the scans of similar areas in IFNα-C treated mice showed 4.2 ± 2 cells (p = 0.01, n = 8). Fundoscopic examination of PBS-treated mice showed perivascular deposits, engorged blood vessels, and hemorrhage (**Fig 6B**), while the eyes from IFNα-C treated mice did not exhibit similar damage. Fourteen days after immunization, eyes were harvested, fixed and stained with hematoxylin and eosin (**Fig 6C**). Infiltrates of inflammatory cells in retina and vitreous were observed, which is consistent with the observation from SD-OCT scans. We also noted retinal buckling and increased number of nuclei within the inner plexiform layer in the control-treated eyes. The IFNα-C treated eyes, in contrast, were protected from the damage. To evaluate the course of the disease, we employed a clinical score: 0, no disease; 1, numerous infiltrating cells; 2, 1 + engorged blood vessels; 2.5, 2 + hemorrhage and deposits; 3, 2.5 + retinal edema and detachment. The clinical data, including the histology, fundus images, and OCT b-scans were evaluated by 3 masked reviewers who evaluated 5 mice for each treatment at each interval. Their values were averaged for each mouse at each time point. Statistical significance at each interval was determined using the Mann-Whiney test.

Solvent treated mice had reached a clinical score of 2.7 ± 0.45 on day 14, while the IFNα-C treated mice were at 0.4 ± 0.4 (n = 8) (**Fig 6D**), suggesting the protective effect of IFNα-C against the development of EAU. Although, treatment with IFNα-C was stopped at 3 week, no relapse of the disease symptoms was noted up to 5 weeks when the study was terminated.

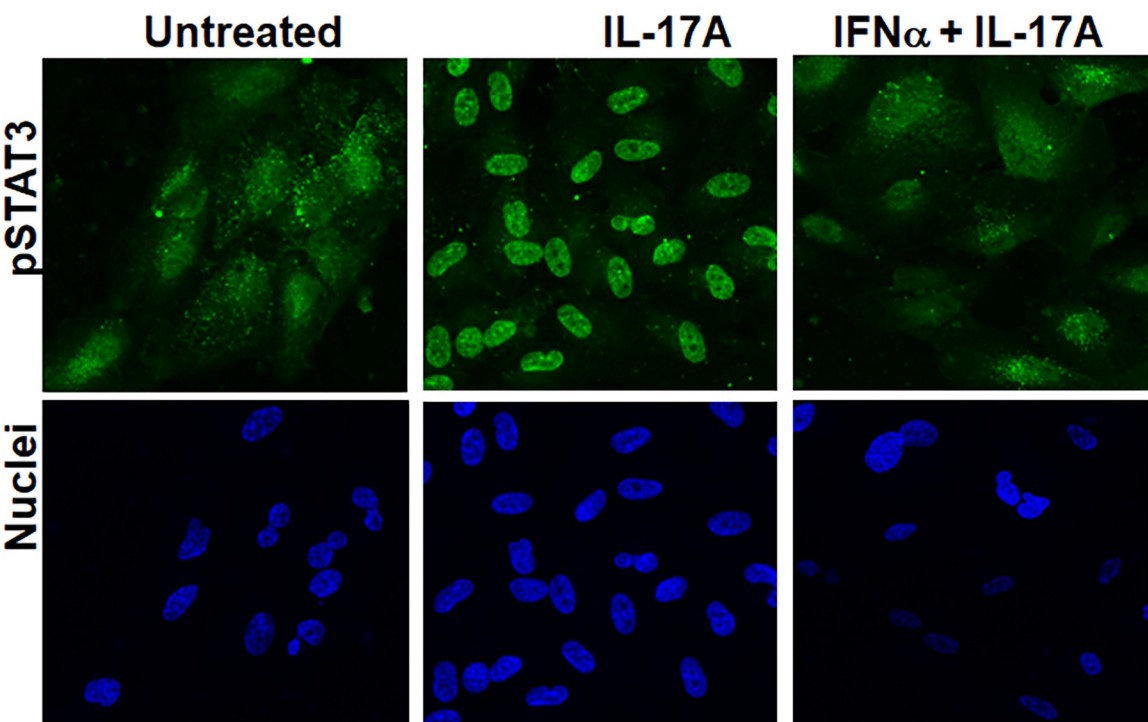

**Fig 3. IFNα-C prevented signaling from IL-17A.** ARPE-19 cells were seeded on 8 well microscopic slides and grown overnight. They were placed in low serum medium and treated with IFNα-C (3 μM) for 2 hr followed by treatment with IL-17A (50 ng/ml) for 30 min. Cells were fixed, permeabilized with 1% Triton X-100 (in PBS) and stained with an antibody to pSTAT3, followed by staining with Alexa 488-conjugated secondary antibody and DAPI and imaged in a fluorescence microscope at 40x magnification.

To test if the structural damage seen above affected retinal function, we conducted electro-retinography (ERG) mice prior to immunization and 4 weeks after the immunization and treatment (Fig 7).

Control mice, treated only with PBS, showed 50–60% decline in amplitudes across all intensities in a- and b- waves in both dark-adapted (scotopic) and light-adapted (photopic) conditions. A two-way ANOVA followed by post-hoc Sidak test to identify differences between groups at the same light intensity gave a p value of 0.01 and <0.0001 at -10 and 0 dB, respectively. In contrast to this, mice that were treated with IFNα-C showed a modest decline in scotopic ERG amplitudes that did not reach statistical significance, suggesting the preservation of retinal function by treatment with IFNα-C.

## IFNα-C inhibits IRBP-induced *ex vivo* proliferation of splenocytes from EAU mice

To evaluate the effect of IFNα-C on peripheral immune system, splenocytes from mice four weeks after immunization and treatment with IFNα-C were cultured and their proliferation in response to IRBP was tested in the presence or absence of IFNα-C. As shown in Fig 8, treatment of splenocytes with IRBP induced their proliferation, which was significantly inhibited when IFNα-C was simultaneously present. This result is consistent with the immunomodulation and therapeutic efficacy noted above.

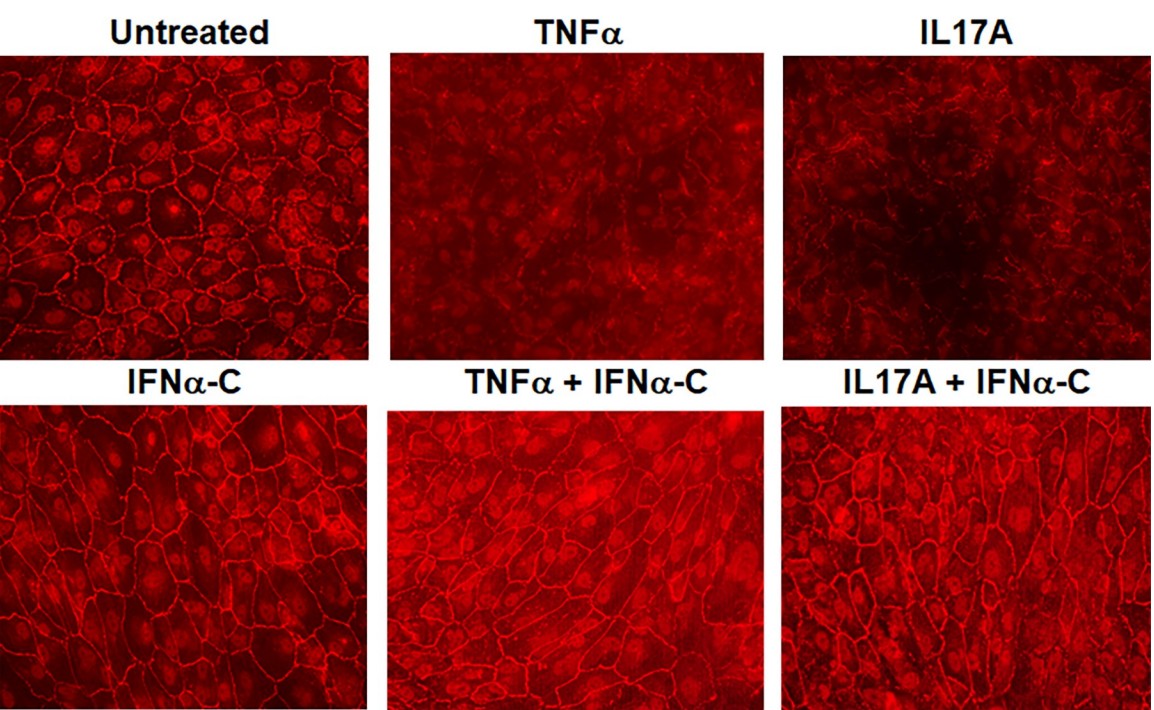

**Fig 4. IFNα-C prevented the loss of tight junctions caused by TNFα or IL-17A.** ARPE-19 cells grown in 8 well slides in low serum for 4 weeks were pre-treated with IFNα-C (3 μM) for 4 hr followed by TNFα or IL-17A (both at 50 ng/ml) for 48 hr. These were compared to untreated cells and cell treated with TNFα alone. Cells were fixed, permeabilized and stained with an antibody to ZO-1, followed by staining with Cy3-conjugated secondary antibody, and imaged by fluorescence microscopy at 40x magnification.

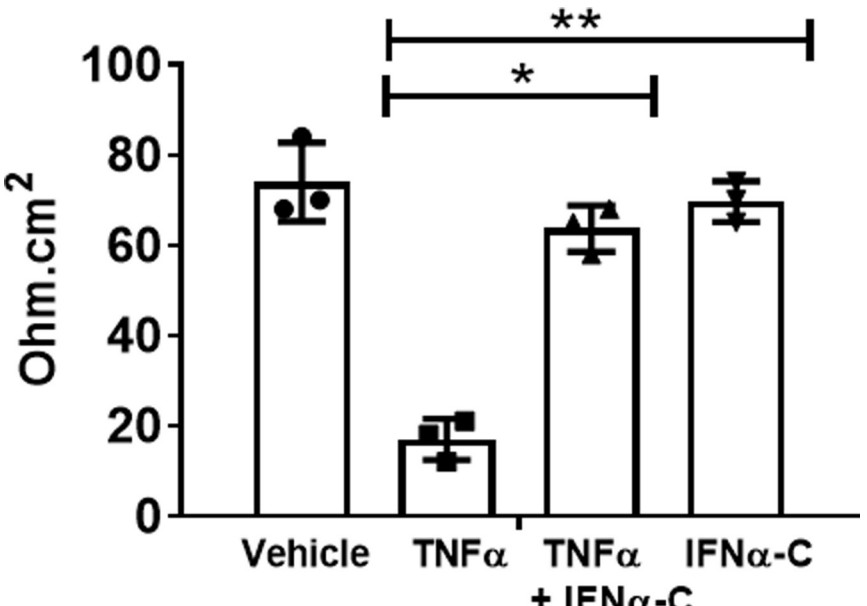

**Fig 5. IFNα-C prevents the reduction of transepithelial resistance caused by TNFα.** ARPE-19 cells grown in 24 well transwell plates in low serum medium for 4 weeks were treated with IFNα-C (3 μM) for 4 hr followed by treatment with TNFα (50 ng/ml) for 48 hr. Untreated cells or those treated with TNFα alone were included alongside. TEER, in individual cells in triplicate was measured using EVOM2 voltohmmeter. The error bars indicate standard deviation. [*], p = 0.001; [**], p = 0.009.

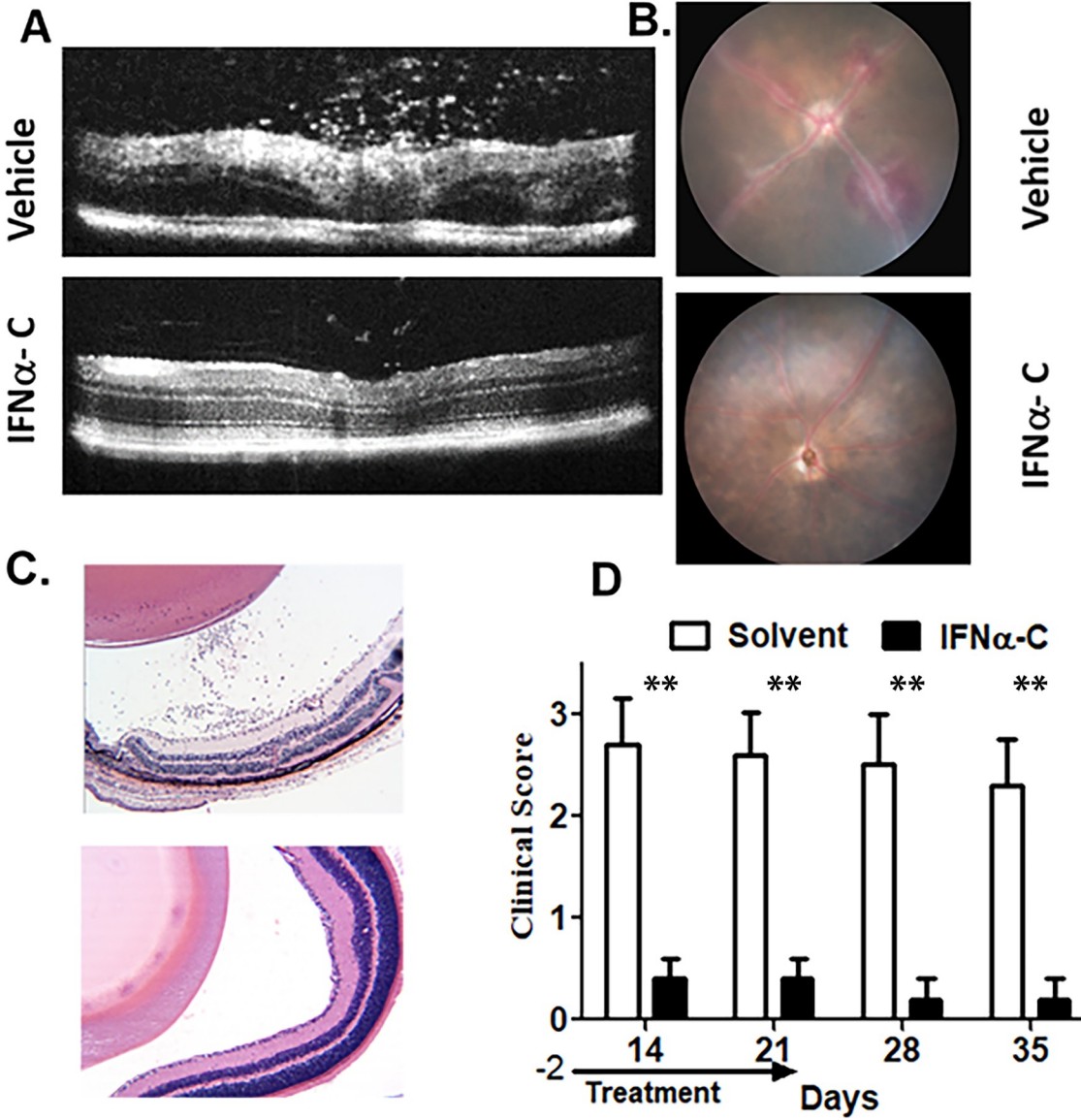

**Fig 6. Protection against EAU by oral gavage with IFNα-C.** Eight-week old female B10.RIII mice (n = 8) were gavaged with IFNα-C (200 μg/mouse) or solvent (PBS) in 200 μl on days -2, -1 and 0. On day 0, mice were immunized with IRBP peptide emulsified in complete Freund's adjuvant. Treatment with IFNα-C or solvent was continued for 3 weeks. **A**. Optical coherence tomography on day 14. Mice treated with solvent showed influx of inflammatory cells and swelling of retina, which were not seen in mice treated with IFNα-C. **B**. Fundoscopic images on day 14 show engorged blood vessels and hemorrhage in solvent treated mice. **C**. Hematoxylin and eosin staining of eyes harvested on day 14 after immunization. Infiltrating cells and retinal folding seen in solvent were not observed in mice treated with IFNα-C. Images were taken at 40x magnification. **D**. A clinical score, as described in Materials and Methods was used. Mice treated with IFNα-C were protected against the symptoms of EAU. The error bars represent standard deviation. The symptoms did not develop in the following two weeks after termination of treatment. **, $p < 0.01$.

## Discussion

Recombinant type I interferons (IFNα and IFNβ), have been approved for the treatment of a number of malignancies, viral infections and multiple sclerosis [46]. In addition, IFNα has been used in Europe to treat various forms of uveitis [27,47]. However, their use in the clinic is associated with severe toxicity, including lymphopenia, depression and weight loss. As an alternative, we have developed an IFNα mimetic from its C-terminal, denoted as IFNα-C that

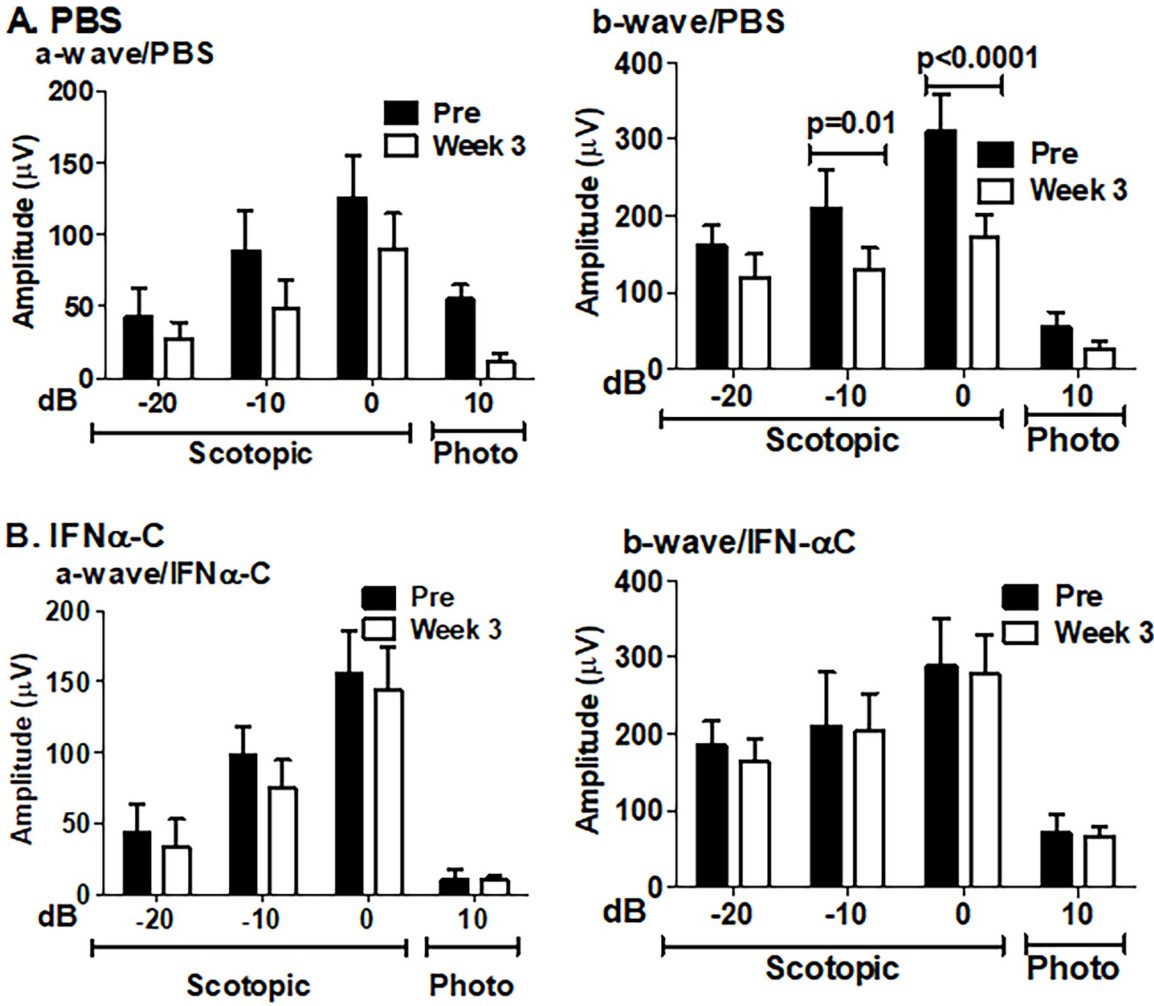

**Fig 7. IFNα-C prevented the loss of ERG a- and b- wave amplitudes in EAU mice.** Dark adapted and light adapted electroretinograms were recorded in B10.RIII mice before immunization (pre) and 3 weeks after immunization with IRBP and treatment with solvent (top row) or IFNα-C (bottom row). Average a-wave and b-wave amplitudes are shown. The error bars represent standard deviation. The same cohort of mice were used for all data points. P values were determined by 2-way ANOVA using the Sidak method for multiple comparisons. N = 8.

offers two distinct advantages: 1) In a mouse model of MS, IFNα-C protected mice against the remitting/relapsing episodes of paralysis, without the attendant toxicity seen in the parent IFN [30]; 2) Since receptors for type I IFN are ubiquitous, therapeutic IFN is often "soaked up" by the unwanted cells and tissues before reaching its target organ, which may explain why a higher dose is required to attain therapeutic efficacy. The uptake of interferon by undesired tissues may contribute to its toxicity. We have shown previously that the higher the affinity of the type I IFN binding to its receptor, the greater is its toxicity. For example, IFNα2 bound to its receptor with 10-fold higher affinity than the non-toxic IFNτ [48]. Since the IFNα-C peptide acts independently of binding to the extracellular domain of its receptor, it is conceivable that this property makes it less toxic [49]. In a series of experiments carried out over twenty years, we have demonstrated for both type I and type II IFNs that the N-terminus of the ligand interacts with the extracellular domain of its cognate receptor and determines the species specificity of IFN action, while the C terminus, after endocytosis, binds to the intracellular domain of the receptor and initiates JAK/STAT signaling similar to the parent IFN (reviewed in [49,50]).

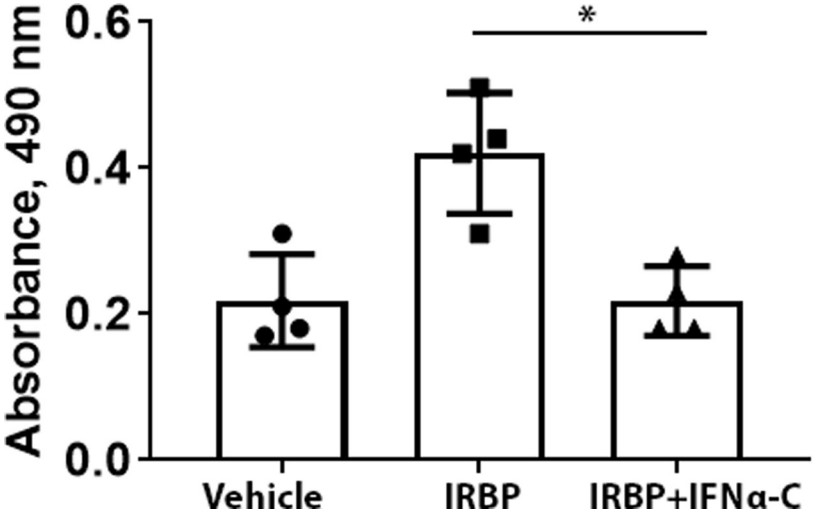

**Fig 8. IFNα-C suppressed the IRBP-induced proliferation of splenocytes.** Splenocytes were harvested from IRBP-immunized B10.RIII mice 3 weeks after immunization. Splenocytes ($5x10^5$ cells/well) in 96 well plates were grown in RPMI with 10% FBS. IFNα-C (3 μM) was added for 4 hr followed by addition of IRBP (50 μg/ml) and cells were grown for 72 hrs. Cell proliferation was measured as described in Material and Methods. The error bars indicate standard deviation. P = 0.02.

This model was further tested with the poxvirus decoy receptors for both type I and type II IFNs that are secreted and code only for the extracellular domain of the receptor. The C-terminal peptides from type I or type II IFNs bypassed these decoy receptors and protected mice against lethal dose of vaccinia virus (reviewed in [51,52]). Furthermore, IFNα-C was shown to phosphorylate tyrosine kinase TYK2 and the transcription factor STAT1 in WISH cells [30], confirming the ability of these peptides to recruit the same signaling molecules within the cell as the parent IFN. In future studies, intravitreal delivery of IFNα-C peptide will be investigated to allow more of the effector molecule become available where it is needed, without losing it *en route* and protecting the individual from toxic side effects.

In this work, we have shown that IFNα-C increases the expression of the transcription factor Foxp3. Foxp3 is required for the generation of Tregs that suppress immune response [38]. In addition, type I IFN can increase the functional activity of regulatory T cells by enabling the conversion of conventional T cells into regulatory T cells. Although, the induction of Foxp3 here was observed in ARPE-19 cells, its effects are most relevant in regulatory T cells. Type I IFN has also been shown to cause the polarization of macrophages into the M2 subtype that have a neuroprotective role [38]. Elevated production of TGFβ as recorded above also contributes to reducing the inflammatory response.

An emerging theme in the regulation of cytokine levels is the production of the following set of proteins, tristetraprolin (TTP), and Twist 1 and 2 that bind to the AU-rich region (ARE) in the 3'-UTR of cytokine mRNAs and cause their degradation. Examples of regulation at the level of mRNA degradation include: TNFα [53], IL-2 [54], IL-6 [55], IL-12 [56], IL-23, IFNγ [57], Ccl2, and Ccl3 [36]. In **Table 2**, we have demonstrated the ability of IFNα-C to enhance TWST1 and TTP synthesis that are likely to down-regulate the levels of inflammatory cytokines by causing degradation of the corresponding mRNAs.

The retinal pigment epithelium constitutes the outer blood retinal barrier (BRB). Since enhanced levels of TNFα have been noted in uveitis patients, we have demonstrated that IFNα-C counteracts the effect of TNFα by down-regulating the cytokines and chemokines

that are induced by TNFα (**Table 3**). IL-8, the chemokine that recruits neutrophils and leuko-cytes to the site of inflammation, is also inhibited by type I IFN [58]. Type I IFNs exert anti-inflammatory effects by using a variety of both STAT-dependent and independent mecha-nisms that contribute to resolution of inflammation. Acting via JAK/STAT pathway, and using a complex consisting of STAT1/STAT2/IRF9 (ISGF3), a number of genes are activated, includ-ing suppressor of cytokine signaling 1 (SOCS1). SOCS1 binds to Rac1, an intracellular GTPase, ubiquinylates and degrades it, which leads to a reduction in the generation of reactive oxygen species (ROS) and suppression of the NLRP3 inflammasome [59–61]. The presence of inflam-matory substances (e.g., complement) and the oxidation products from drusen and other regions in the eye initiate signals for their activation leading to the production of IL-1β and IL-18, resulting in pyroptosis [22]. Membrane rupture from pyroptosis leads to the release of IL-1β, IL-18 and the cellular oxidation products, thus enhancing chemotactic migration of T helper cells and antigen presenting cells. Aside from its role in suppression of NLRP3, SOCS1 (induced by IFN) also acts by binding to the activation domain of JAK2 and TYK2 and the adaptor protein MyD88 adaptor like (MAL), thus limiting the extent and duration of inflam-matory responses eliciting from IL-6 [62], IFNγ[62,63], and MyD88 [62]. The end result is the prevention of pathological consequences of uncontrolled expression of inflammatory signals, some examples of which were noted in **Table 3**.

We have shown the ability of IFNα-C to suppress signaling arising from NF-kB activation. Suppression of NF-kB leads to the downregulation of inflammatory cytokines such as IL-6 and TNFα. In addition, IL-17 production as well as its downstream signaling is suppressed by type I IFN [41, 42, 64], which protects against the damage caused by IL-17 in several autoimmune disorders. We showed that IFNα-C suppressed signaling from IL-17A by blocking activation of STAT3. Type I IFN can also downregulate IFNγ, a product of Th1 cells, by binding to its 3'-UTR and degrading its mRNA [57]. In addition, SOCS 1/3 induced by IFNα is likely to inhibit Th1 response. TGFβ and IL-6 contribute to the polarization of Th17 cells [65, 66], while IL-10 has anti-inflammatory properties and protects against EAU [67]. Either Th1 or the Th17 response may cause autoimmune multiple sclerosis and uveitis [68]. However, blockade of the Th1 response leads to an elevated Th17 response, while a deficiency of Th17 response increases the Th1 response [69]. To effectively treat autoimmune disease, inhibition of both Th1 and Th17 responses is critical. IFNα-C achieves this dual inhibition.

TNFα and IL-17A have been shown to disrupt blood-brain barrier (BBB) by destabilizing tight junctions [45]. We showed that IFNα-C was able to prevent the disruption of ZO-1 distri-bution on ARPE-19 cells and consequently to counteract the reduction of transepithelial elec-trical resistance (**Figs 4 and 5**). Having confirmed the ability of IFNα-C to reduce inflammatory response and to protect against damage to cell permeability properties in ARPE-19 cells, we proceeded to test it ability to protect B10.RIII mice against the development of autoimmune uveitis. These beneficial effects observed on ARPE-19 cells may have contributed to the protection afforded by IFNα-C in mice with EAU, including the prevention of influx of inflammatory cells, and preservation of retinal structure and function.

Cell penetrating peptides (CPPs) offer an attractive alternative to gene therapy approaches [70]. Large amounts of peptides can be generated by chemical synthesis or by bacterial produc-tion of recombinant peptides. Attachment of basic amino acids (nine arginines, R9), or conju-gation with palmitoyl-lysine allows penetration across plasma membrane. Alternatively, a peptide sequence targeting a specific tissue can be generated by attaching a sequence unique to the target tissue. We have shown previously that a R9-conjugated peptide from SOCS1 kinase inhibitor region (R9-SOCS1-KIR), used topically was effective in protecting against EAU, both prophylactically and therapeutically [31]. Palmitoyl-lysine conjugated SOCS1-KIR was reported by others as effective in preventing EAU [71, 72]. Eye drop formulations of other

CPPs, vascular endothelial growth inhibitor (VEGI) [73], pigment epithelium derived factor peptide (PEDF) peptide [74, 75], and endostatin [76] have been reported.

We delivered IFNα-C by oral gavage in these experiments. We have previously shown the therapeutic efficacy of orally administered type II IFN peptide, IFNγ(95–132) in protecting against a lethal dose of vaccinia virus [34]. In earlier work we demonstrated the ability of palmitoyl-lysine conjugated to type II IFN peptide to carry the peptide across the plasma membrane by fusing it with FITC and testing in cells in culture and in mouse peritoneal cells by fluorescence microscopy [77]. However, we have not yet conducted detailed pharmacokinetic experiments of this cell-penetrating protein, and cannot comment about its stability or biodistribution in mice. Orally administered effectors of serotonin receptor were shown by us to protect retinal structure and function in a mouse model of macular degeneration [78.79]. These examples suggest the oral bioavailability of these drugs through recognition by the mucosal immune system. Also, CPP for SOCS1-KIR given i.p. was shown to cross blood-brain barrier to protect against severe form of remitting/relapsing experimental allergic encephalomyelitis (EAE), the mouse model of multiple sclerosis (MS) [62]. Systemic delivery of these peptides is likely to have therapeutic effect in CNS or retinal disorders.

## Conclusion

IFNα-C had a protective role in affording an immunosuppressive environment as well in preventing damage caused by the relevant inflammatory cytokines to ARPE-19 cells. These protective properties may have contributed to the preservation of structural and functional properties during the course of autoimmune uveitis in mice. It is conceivable that these properties will also be helpful in treating other inflammatory eye diseases, such as age related macular degeneration (AMD) and autoimmune keratitis.

## Supporting information

**S1 Checklist. The ARRIVE guidelines checklist animal research: Reporting *in vivo* experiments.**
(PDF)

## Acknowledgments

This project was supported by the Shaler Richardson Professorship Endowment, which had no role in study design, data collection and analysis, decision to publish, or preparation of the manuscript. The University of Florida been awarded a patent (US 9,951,111B2) governing the interferon peptide mimetic used in this study. Two of the authors (CMA and HMJ) may obtain royalties from the licensing of this technology.

## Author Contributions

**Conceptualization:** Chulbul M. Ahmed.

**Formal analysis:** Chulbul M. Ahmed, Cristhian J. Ildefonso, Alfred S. Lewin.

**Investigation:** Chulbul M. Ahmed.

**Supervision:** Howard M. Johnson, Alfred S. Lewin.

**Visualization:** Cristhian J. Ildefonso.

**Writing – original draft:** Chulbul M. Ahmed.

**Writing – review & editing:** Cristhian J. Ildefonso, Howard M. Johnson, Alfred S. Lewin.

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
