## [Decision Letter · Decision Letter 0]

24 Jan 2020

PONE-D-19-33376

A Cell Penetrating Peptide from Type I Interferon Protects the Retina in a Mouse Model of Autoimmune Uveitis

PLOS ONE

Dear Dr. Lewin,

Thank you for submitting your manuscript to PLOS ONE. After careful consideration, we feel that it has merit but does not fully meet PLOS ONE’s publication criteria as it currently stands. Therefore, we invite you to submit a revised version of the manuscript that addresses the points raised during the review process.

We would appreciate receiving your revised manuscript by Mar 08 2020 11:59PM. To enhance the reproducibility of your results, we recommend that if applicable you deposit your laboratory protocols in protocols.io, where a protocol can be assigned its own identifier (DOI) such that it can be cited independently in the future. For instructions see: http://journals.plos.org/plosone/s/submission-guidelines#loc-laboratory-protocols

We look forward to receiving your revised manuscript.

Kind regards,

Anand Swaroop

Academic Editor

PLOS ONE

Journal Requirements:

2. Please complete and submit a copy of the ARRIVE Guidelines checklist, a document that aims to improve experimental reporting and reproducibility of animal studies for purposes of post-publication data analysis and reproducibility: https://www.nc3rs.org.uk/arrive-guidelines. Please include your completed checklist as a Supporting Information file. Note that if your paper is accepted for publication, this checklist will be published as part of your article.

Specifically, please ensure that you revise your methods section to include the method of euthanasia and any anaesthesia used, as well as how frequently the condition of the animals was monitored.

"ASL Shaler Richardson Endowment https://research.ufl.edu/ufrf.html. The sponsors played no role in the study design."

5. We note that you have a patent relating to material pertinent to this article. Please provide an amended statement of Competing Interests to declare this patent (with details including name and number), along with any other relevant declarations relating to employment, consultancy, patents, products in development or modified products etc. Please confirm that this does not alter your adherence to all PLOS ONE policies on sharing data and materials, as detailed online in our guide for authors http://journals.plos.org/plosone/s/competing-interests by including the following statement: "This does not alter our adherence to  PLOS ONE policies on sharing data and materials.” If there are restrictions on sharing of data and/or materials, please state these. Please note that we cannot proceed with consideration of your article until this information has been declared.

Reviewers' comments:

Reviewer's Responses to Questions

**Comments to the Author**

1. Is the manuscript technically sound, and do the data support the conclusions?

Reviewer #1: Partly

Reviewer #2: Yes

2. Has the statistical analysis been performed appropriately and rigorously? 

Reviewer #1: Yes

Reviewer #2: No

3. Have the authors made all data underlying the findings in their manuscript fully available?

Reviewer #1: Yes

Reviewer #2: Yes

4. Is the manuscript presented in an intelligible fashion and written in standard English?

Reviewer #1: Yes

Reviewer #2: Yes

5. Review Comments to the Author

Reviewer #1: On Figure 1, the three rightmost bars have the same label "TNF" on them. Please fix.

The phrase "cell penetrating peptide" may be misleading. It could be that the lipo-IFNalpha1 (152-189) peptide enters the cell by engulfment, endocytosis, or some other mechanism. There are no data cited or experiments in the manuscript that speak to the mechanism of cell entry of this specific peptide. A re-phrasing of the term is warranted.

A more appropriate "negative control" for the anti-inflammatory "Human lipo-IFNalpha1 (152-189) peptide" is a shuffled peptide sequence in the same DMSO vehicle. A qualifier is needed. Is palmitate optimal or do other lipophilic groups work as well? Please cite literature or qualify.

The authors should disclose more information on how the human lipo-IFNalpha1 (152-189) peptide does its thing. The authors should comment on any amino acid sequence differences between mouse and the human IFNalpha1 (152-189) peptide. A comment on the sequence conservation among mammals or vertebrate would be helpful to the reader, too. Specifically, this sentence fragment needs expansion or clarification: "As improved versions of the IFNalpha-C become available, ...."

How dosage of the peptide was arrived at is not obvious. Please explain how and why the doses were selected.

It is not clear if the lipo-IFN-peptide survives the gavage process intact. IE, it presumably is exposed to digestive processes and needs to pass through the gut epithelium in order to interact with T-cells and the like. Comment on this.

The ARPE-19 cell work is interesting and may shed light on the mechanism. However, the state of differentiation is probably "partial". Are there melanosomes? How much Rpe65 is expressed relative to actual human RPE cells? Some qualification is needed.

The use of the clinical grading scale (Fig 6D) is accepted practice in evaluating inflammation in the mouse retina caused by the IRBP160-181 peptide (clarify if human, mouse, or same sequence?) or other uveitogenic peptides. However, the authors did not disclose if the histo-pathologic slides of samples were presented to the graders (and clarify how many graders there were) in a double blinded manner. Please use the Mann-Whitney U or the Kruskal-Wallis H post hoc tests on the ranked "clinical" data.

Reviewer #2: The authors reported the effects of cell penetrating INFa in RPE cells and a autoimmune uveitis mouse model. The results are promising. Here are my comments:

1. In the main text of Table 2, the RPE cells had received "inflammatory stimulus". However, other than treated with low serum medium, the "inflammatory stimulus" was not clear.

2. There was no statistics in Table 3.

3. In Figure 1, all 3 treated groups were labelled as TNFa. And no indication of statistical comparison between the bars.

4. In Figure 2, the background noise of the NF-kB staining was too high.

5. In all the immunofluorescence staining, quantification with statistics are desired.

6. On page 14, the sentences "These were added to differentiated ARPE-19 cells for 48 hr at 50 ng/ml each. Treatment

here was for 48 hr as opposed to 24 hr for freshly cultured cells, because it takes longer for the differentiated cells to show the damage." could be re-written in standard English.

6. PLOS authors have the option to publish the peer review history of their article (what does this mean?). If published, this will include your full peer review and any attached files.

Reviewer #1: No

Reviewer #2: Yes: WAI KIT CHU

---

## [Author Response · Author response to Decision Letter 0]

3 Feb 2020

Response to reviewers 01/31/20

We would like to thank the reviewers for their careful reading of the original submission and their useful comments, which should result in an improved paper.

Editorial Comments:

Specifically, please ensure that you revise your methods section to include the method of euthanasia and any anaesthesia used, as well as how frequently the condition of the animals was monitored.

On page 9 (our numbering system) we added the statements: “The health of the mice was monitored daily.” and “For these procedures mice were anesthetized with a mixture of ketamine (95 mg/kg) and xylazine (5-10 mg/kg), and at the conclusion of the study mice were euthanized by inhalation of CO2 followed by cervical dislocation.”

Please state what role the funders took in the study. 

In the Acknowledgements we now state: “This project was supported by the Shaler Richardson Professorship Endowment, which had no role in study design, data collection and analysis, decision to publish, or preparation of the manuscript.”

Also, in the Acknowledgements, we now state: The University of Florida been awarded a patent (US 9,951,111B2) governing the interferon peptide mimetic used in this study. Two of the authors (CMA and HMJ) may obtain royalties from the licensing of this technology.

The “data not shown” in the text has been modified to cite our previous publication that used the control antibody described. 

5. Review Comments to the Author

Reviewer #1: On Figure 1, the three rightmost bars have the same label "TNF" on them. Please fix.

Response: Thanks. This was truncated during pasting, and has now been corrected. (This comes from letting the senior author submit the manuscript!)

The phrase "cell penetrating peptide" may be misleading. It could be that the lipo-IFNalpha1 (152-189) peptide enters the cell by engulfment, endocytosis, or some other mechanism. There are no data cited or experiments in the manuscript that speak to the mechanism of cell entry of this specific peptide. A re-phrasing of the term is warranted.

Response: Thank you for requesting this clarification. The mechanism of lipidated peptides entry via lipid microdomains has been well characterized (e.g., PMID: 12165521, 17372016). However, to avoid confusion we have taken out “cell penetrating peptide” from the title and called it “a C-terminal peptide”.

A more appropriate "negative control" for the anti-inflammatory "Human lipo-IFNalpha1 (152-189) peptide" is a shuffled peptide sequence in the same DMSO vehicle. A qualifier is needed. Is palmitate optimal or do other lipophilic groups work as well? Please cite literature or qualify.

Response: In our previous publication by Ahmed et al (J. Interferon Cytokine Res 34: 802-809; 2014; ref 30 in the manuscript), we had described a scrambled peptide for IFNα1(152-189), which was devoid of any IFN-related antiviral activity, or protection of mice against experimental autoimmune encephalomyocarditis EAE). Based on this, we have chosen to use the solvent in this manuscript. In place of palmitoyl-lysine, we have used a cationic sequence, polyarginine (R9) for cell penetration of therapeutic peptides, with equal success. 

The authors should disclose more information on how the human lipo-IFNalpha1 (152-189) peptide does its thing. The authors should comment on any amino acid sequence differences between mouse and the human IFNalpha1 (152-189) peptide. A comment on the sequence conservation among mammals or vertebrate would be helpful to the reader, too. Specifically, this sentence fragment needs expansion or clarification: "As improved versions of the IFNalpha-C become available, ...."

Response: Thank you for requesting this clarification. In our previous publication by Ahmed et al (J. Interferon Cytokine Res 34: 802-809; 2014; ref 30, mentioned in the Introduction and described in more detail in the Discussion), we have shown that INFα-C peptide acts through phosphorylation of TYK2 and STAT1α. Thus, it acts through the JAK/STAT pathway similar to parent interferon. Detailed studies from the laboratory of Howard Johnson (an author on this manuscript) have delineated the functional role of structural motifs in IFN and its receptor, where the N-terminus was found to bind to the extracellular part of the IFN receptor, while the C-terminus of IFN interacts with the cytoplasmic domain of IFN receptor to recruit JAK/STAT pathway components to provide nearly all the biological activities found in the parent IFN.

Between the regions 152-189, there is 70% homology between human and mouse IFNα1 sequences, most of which are conservative differences. There is 90% homology between the sequences if the conservative differences are not counted. Type I interferons share a 30-85% homology within a species. 

On page 4 we add the statement: “Between the regions 152-189, there is 70% homology and 90% conservation between human and mouse IFNα1 sequences.” 

We have deleted the sentence, “As improved versions…”. The last sentence in the first paragraph on page 20, now reads as follows: “In future studies, intravitreal delivery of IFNα-C peptide will be investigated to allow more of the effector molecule become available where it is needed, without losing it en route and protecting the individual from toxic side effects.”

How dosage of the peptide was arrived at is not obvious. Please explain how and why the doses were selected.

Response: The peptide dose was based on our previous study where oral gavage of IFN peptide had protected mice against lethal dose of vaccinia virus (Ahmed et al. J. Immmunol. 178: 4576-4583; 2007; ref 34 in the manuscript).

On p. 9, we added the statement: “The peptide dose was based on our previous study in which oral gavage of IFN peptide had protected mice against lethal dose of vaccinia virus (34).”

It is not clear if the lipo-IFN-peptide survives the gavage process intact. IE, it presumably is exposed to digestive processes and needs to pass through the gut epithelium in order to interact with T-cells and the like. Comment on this.

Response: In our previous study (Ahmed et al. J. Immmunol. 178: 4576-4583; 2007; ref 34 in the manuscript), we delivered the IFN peptide by oral gavage, which protected mice against a lethal dose of vaccinia virus. This would indicate that at least some of the lipophilic peptide survived the intestinal milieu and is able to reach the appropriate cells to provide protection against the virus. There have been several reports in literature, where orally delivered peptides show the therapeutic effect. Insulin, which is the first polypeptide that is mentioned in defense of therapeutic peptides, was given orally to obtain the therapeutic benefit (Rehmani and Dixon, 2018, Peptides doi: 10.1016/j.peptides.2017.12.014). For treatment of CNV, oral delivery of VEGF inhibitory peptide was used (Tarallo V et al., 2020, Int J. Mol Sci 21 (2); doi: 10.3390/ijms2102410). A review on this subject is provided in (Drucker, Nature Rev Drug Discov doi: 10.1038/s41573-019-0053-0). We now cite this review on page 9.

The ARPE-19 cell work is interesting and may shed light on the mechanism. However, the state of differentiation is probably "partial". Are there melanosomes? How much Rpe65 is expressed relative to actual human RPE cells? Some qualification is needed.

Response: Thanks for this remark. On age 8, we added the sentence: “It should be noted that despite their ability to form tight junctions, ARPE-19 monolayers do not form melanosomes and express low levels of marker proteins such as CRALBP and RPE65.” ARPE-19 cells were obtained from the American Type Culture Collection (Manassas, VA) cells from the first culture were frozen immediately. ARPE-19 cells were discarded after three subsequent passages. The cell line was authenticated by STR analysis using GenePrint 10 System from Promega Corporation (Madison, WI, USA) that allows co-amplification of repeat regions of nine short tandem repeat human loci, including the ASN-0002 loci (TH01, TPOX, vWA, Amelogenin, CSF1PO, D16S539, D7S820, D13S317 and D5S818) as well as D21S11. The samples were processed on ABI 3130XL Genome Analyzer. Data were analyzed using GeneMapper® v4.0 software and compared to positive and negative controls.

The use of the clinical grading scale (Fig 6D) is accepted practice in evaluating inflammation in the mouse retina caused by the IRBP160-181 peptide (clarify if human, mouse, or same sequence?) or other uveitogenic peptides. However, the authors did not disclose if the histo-pathologic slides of samples were presented to the graders (and clarify how many graders there were) in a double blinded manner. Please use the Mann-Whitney U or the Kruskal-Wallis H post hoc tests on the ranked "clinical" data.

Response: It was the human IRBP peptide that was used. This has been added on page 9. The clinical data, including the histology fundus images, and OCT b-scans were evaluated by 3 masked (ignorant of treatment group) reviewers who evaluated 5 mice for each treatment at each interval. Their values were averaged for each mouse at each time point. Statistical significance at each interval was determined using the Mann-Whiney test (This is described on p 16.)

Reviewer #2: The authors reported the effects of cell penetrating INFa in RPE cells and a autoimmune uveitis mouse model. The results are promising. Here are my comments:

1. In the main text of Table 2, the RPE cells had received "inflammatory stimulus". However, other than treated with low serum medium, the "inflammatory stimulus" was not clear.

Response: We thank the reviewer for pointing this out. This table was not meant to show a response to inflammation, but rather that treatment with the interferon mimetic peptide induces the expression of genes that can mitigate inflammation. The modified sentence now reads as follows: “ARPE-19 cells were incubated in the presence or absence of IFNα-C peptide,……”

2. There was no statistics in Table 3.

Response: The last column of Table 3 was truncated during preparation of the manuscript. We apologize. The last column with P values that indicate the statistical significance of the values is shown in the Table 3 now. 

3. In Figure 1, all 3 treated groups were labelled as TNFa. And no indication of statistical comparison between the bars.

Response: The replacement figure has the bar and asterisk to indicate the p value. 

4. In Figure 2, the background noise of the NF-kB staining was too high.

Response: The antibody used was for total p65, and not for phosphorylated p65, hence the high background: unphosphorylated p65 is cytoplasmic (please see response to comment 5).

5. In all the immunofluorescence staining, quantification with statistics are desired.

Response: This is an excellent suggestion for future experiments. We have used the immunofluorescence experiments as qualitative measure of the nuclear translocation. For the p65 staining, it was the antibody to p65 and not p-p65. This allowed us to visualize the p65 in the cytoplasm as well as the nucleus. The lack of fluorescence in the nuclei in the untreated and IFNα-C treated cells and the brightness in nuclei of TNF-α treated cells is a qualitative measure of the nuclear translocation of p65 in Figure 2 and lack of translocation of pSTAT3 in Figure 3.

6. On page 14, the sentences "These were added to differentiated ARPE-19 cells for 48 hr at 50 ng/ml each. Treatment here was for 48 hr as opposed to 24 hr for freshly cultured cells, because it takes longer for the differentiated cells to show the damage." could be re-written in standard English.

Response: The second sentence was changed to: “We treated differentiated ARPE-19 cells for 48 hours, because they are highly resistant to damage”

---

## [Editor Report · Decision Letter 1]

5 Feb 2020

A C-Terminal Peptide from Type I Interferon Protects the Retina in a Mouse Model of Autoimmune Uveitis

PONE-D-19-33376R1

Dear Dr. Lewin,

We are pleased to inform you that your manuscript has been judged scientifically suitable for publication and will be formally accepted for publication once it complies with all outstanding technical requirements.

With kind regards,

Anand Swaroop

Academic Editor

PLOS ONE
---

## [Editor Report · Acceptance letter]

13 Feb 2020

PONE-D-19-33376R1 

A C-Terminal Peptide from Type I Interferon Protects the Retina in a Mouse Model of Autoimmune Uveitis 

Dear Dr. Lewin:

I am pleased to inform you that your manuscript has been deemed suitable for publication in PLOS ONE. Congratulations! Your manuscript is now with our production department. 

With kind regards,

on behalf of

Dr. Anand Swaroop 

Academic Editor

PLOS ONE